# Investigation of the Bimodal Leaching Response of RAM Chip Gold Fingers in Ammonia Thiosulfate Solution

**DOI:** 10.3390/ma16144940

**Published:** 2023-07-11

**Authors:** Peijia Lin, Zulqarnain Ahmad Ali, Joshua Werner

**Affiliations:** Department of Mining Engineering, University of Kentucky, 310 Columbia Ave, Lexington, KY 40506, USA; peijia.lin@uky.edu (P.L.); zulqarnain.a.ali@uky.edu (Z.A.A.)

**Keywords:** waste RAM chips, thiosulfate leaching, Au recovery, Ni recovery, decopperization, cementation, copper sulfides, extractive metallurgy

## Abstract

Oxidative thiosulfate leaching using Cu(II)-NH_3_ has been explored for both mining and recycling applications as a promising method for Au extraction. This study seeks to understand the dissolution behavior of Au from waste RAM chips using a Cu(II)-NH_3_-S_2_O_3_ solution. In the course of this work, bimodal leaching and Au loss were observed in a manner that we have not identified in the literature. Identification of the existence of a specific Au-Ni-Cu lamellar structure in the gold fingers from RAM chips by scanning electron microscopy and energy dispersive X-ray spectroscopy (SEM-EDS) revealed the possibility of interference between Au recovery and the existence of Cu and Ni. During leaching, the co-extraction of Ni was found to predict a negative impact on the Au recovery, as a result of chemical interactions from the Au-Ni-Cu interlayer. Decopperization as a pretreatment was found necessary to remove the pre-existing Cu and promote Au leaching. As part of the study parameters, such as Cu(II) concentration, aeration rates, thiosulfate and ammonia concentrations, particle sizes, and temperatures, were investigated. A satisfactory Au recovery of 98% was achieved using 50 mM Cu(II), 120 mL/min aeration rate, 0.5 M (NH_3_)_2_S_2_O_3_, and 0.75 M NH_4_OH (i.e., AT/AH ratio of 0.67) for 4 h residence time at room temperature (25 °C). However, there were several high recoveries prior to Au loss from the lixiviant. It was revealed that the main cause of lower Au recovery was due to a precipitation or cementation reaction that included a sulfur species formation. Because of the bimodal leaching, a composite response comprised of the time to Au loss and maximum recovery was developed, termed leaching proclivity, to facilitate statistical analysis. Furthermore, this study explores the interactions between Au-Ni-Cu and provides suggestions for improving Au thiosulfate leaching under the interference of co-existing metals from waste PCB materials.

## 1. Introduction

Rapid growth in technological innovation has resulted in the ongoing increase in waste printed circuit boards (WPCBs) [1]. It is expected that the global e-waste, including WPCBs, will reach 74 million metric tons by 2030, which has almost doubled in the past 16 years [2]. The proper disposal and recycling of WPCBs have created ongoing environmental and social concerns [3]. Conversely, the significant quantities of valuable metals, such as copper, gold, and silver, contained in WPCBs, make it a promising secondary resource [4]. The high contents of Au and other precious metals, especially in small IT equipment, have become the driving force to make the recycling of WPCBs profitable [5]. Currently, hydrometallurgical methods, such as acid leaching and cyanidation, have been widely used to recover valuable metals from WPCBs. However, the corrosivity of strong acids and the toxicity of cyanidation pose threats to the environment and human society [6]. Alternatively, numerous studies have been devoted to exploring potentially greener chemicals to extract gold from primary and secondary resources. These developed methods include leaching by different types of acids [7,8], thiourea [9], glycine [10,11], and thiosulfate [12,13,14,15]. 

Among the above-mentioned alternatives, thiosulfate leaching has been proven to be an efficient, economic, and greener method to recover Au [16]. One of the merits of thiosulfate leaching for Au from WPCBs is the higher selectivity of thiosulfate ions toward Au, which eliminates the co-extraction of other contaminating metals, such as Fe and Al. Thiosulfate leaching is commonly applied in alkaline conditions, using NH_4_OH as a pH conditioner and Cu^2+^ as an oxidant to enhance the Au recovery. 

The genesis of this work arose from the ideation of addressing the issue of recovering value from e-waste in a manner that has not been attempted. Upon a literature review, the body of work showing the ammoniacal removal of copper was deemed sufficiently interesting to warrant further study. The literature describes the ammoniacal processing of electronic wastes and can be subdivided into leaching [17,18,19], electrowinning [18,20,21,22,23], and electrolyte purification via solvent extraction [24]. As unpressurized ammoniacal copper solutions do not leach gold, a synergistic chemistry was identified to minimize the processing expense in transferring the solid from the copper-centric leaching circuit to the gold-centric leaching circuit. In this, the Cu(II)-NH_3_-S_2_O_3_ system was selected because the complementary utilization of copper in both processes would preclude completely removing the copper lixiviant from the copper-leached material. Furthermore, the correct concentration of oxidized Cu(II) is beneficial to the Au leaching, and as such, oxidation between the copper and gold leaching steps may be beneficial. For convenience, the system is represented in Figure 1, showing the anodic and cathodic reactions in an electrochemical system [25]. 

From Figure 1, the anodic reaction, Au dissolution occurs as Au^0^ is oxidized to Au^+^ by Cu(II) and complexes with NH_3_ to form a stable Au–amine compound (Equations (1) and (2)). In the presence of S_2_O_3_^2−^, leached Au further complexes with S_2_O_3_^2−^ because of the higher affinity between Au^+^ and S_2_O_3_^2−^ (Equation (3)). In the cathodic reaction, the Cu(II)-amine complex (Cu(S_2_O_3_)_3_^5−^) is reduced to Cu(I) and forms a Cu(I)-thiosulfate compound (Cu(S_2_O_3_)_3_^5−^) (Equation (4)). The overall reaction of Au leaching in a Cu(II)-NH_3_-S_2_O_3_ system can be expressed as Equation (5). 

Anodic reaction:(1)$$\mathrm{Au}\to {\mathrm{Au}}^{+}+e^{-}$$
(2)$${\mathrm{Au}}^{+}+2{\mathrm{NH}}_{3}\to \mathrm{Au}{\left({\mathrm{NH}}_{3}\right)}_{2}{}^{+}$$
(3)$$\mathrm{Au}{\left({\mathrm{NH}}_{3}\right)}_{2}{}^{+}+2S_{2}O_{3}{}^{2-}\to \mathrm{Au}{\left(S_{2}O_{3}\right)}_{2}{}^{3-}+2{\mathrm{NH}}_{3}$$

Cathodic reaction:(4)$$\mathrm{Cu}{\left({\mathrm{NH}}_{3}\right)}_{4}{}^{2+}+3S_{2}O_{3}{}^{2-}+e^{-}\to \mathrm{Cu}{\left(S_{2}O_{3}\right)}_{3}{}^{5-}+4{\mathrm{NH}}_{3}$$

Overall reaction: (5)$$\mathrm{Au}+5S_{2}O_{3}{}^{2-}+\mathrm{Cu}{\left({\mathrm{NH}}_{3}\right)}_{4}{}^{2+}\to \mathrm{Au}{\left(S_{2}O_{3}\right)}_{2}{}^{3-}+\mathrm{Cu}{\left(S_{2}O_{3}\right)}_{3}{}^{5-}+4{\mathrm{NH}}_{3}$$

As the chemistry of Au leaching and its exaction mechanism in the Cu(II)-NH_3_-S_2_O_3_ system has been studied extensively over the last 20 years [26,27,28,29], the gap from the literature appears to lie in the processing of e-waste using the combined system described previously. As is well known, Cu and Cu sulfide species can be problematic as cementing agents, causing the deposition of leached Au ions in thiosulfate solutions [30,31,32,33]. In support of this hypothesis, the research by Jeon et al. on Au cementation in the solution containing thiosulfate and PCB materials indicates that the dissolved Au thiosulfate complex is preferentially cemented on the surface of Cu [34,35]. In particular, Au cementation on Cu was significantly enhanced when Al coexisted. To minimize the interference of co-existing metals in PCBs, such as Cu, Ni, Al, and Fe, many researchers have developed pretreatment methods prior to Au thiosulfate leaching.

Table 1 summarizes the literature on Au recovery from waste PCBs using thiosulfate with assistance from an oxidizer. As shown, the reported Au recovery varied significantly, with and without the implementation of physical and chemical pretreatment. Sulfuric acid and H_2_O_2_ was adopted in most of the chemical pretreatments. According to the results presented by Oh et al., using 2 M H_2_SO_4_ and 0.2 M H_2_O_2_, they were able to efficiently remove Cu, Ni, Zn, Fe, and Al and achieved 95% of Au recovery in the subsequent thiosulfate leaching [36]. However, the adoption of strong acid leads to a large reagent consumption by the pH conditioning from acidic to alkaline solutions [36,37,38]. As such, an effective alkaline lixiviant was needed in the pretreatment process to remove Cu. The solubility of Cu in ammoniacal solutions was reported by Koyama et al. [18]. This showed the promise of using a Cu(II)-ammoniacal solution as a potential pretreatment of Cu removal with a synergistic effect between ammoniacal and thiosulfate solutions. 

From the literature, there exists a specific gap with regard to ammoniacal copper removal pretreatment and thiosulfate leaching. This study seeks to determine the effects of various parameters on end-of-life RAM chips on materials prepared with an ammoniacal pretreatment. Specifically, this study provides an understanding of the interaction between Au and other dissolvable metals (particularly Ni) in thiosulfate leaching after a copper leaching pretreatment. The independent variables considered are decopperization, aeration, initial Cu(II) concentrations, ammonium thiosulfate and ammonium hydroxide concentrations, particle sizes, and temperatures with the dependent variables being Au and Ni recovery with time, time until Au precipitation, post-leaching solids composition, and leachate composition. Additionally, SEM-EDS characterization was performed on pre and post-leached materials to better understand the effects of leaching and to assist in explaining the results. 

It is anticipated that this work will provide a deeper understanding of the factors related to Au dissolution from WPCB materials by Cu(II)–ammoniacal thiosulfate leaching with ammoniacal decopperization pretreatment. Specific outcomes include:The effects of a pretreatment to eliminate the thiosulfate consumption by the pre-existing Cu; Effects of control on the oxidation state (both the initial Cu(II) concentration and the aeration to maintain the Cu(II) concentration);Evaluation of the impact of other leachable metals (i.e., Ni and Cu) on Au dissolution; Investigation of other processing factors on Au dissolution, accessed by inhibiting the negative impact by co-extracted metals. 

## 2. Materials and Methods

### 2.1. Materials and Pretreatment

The feed materials used in this study were waste RAM chips, obtained from the recycling services at the University of Kentucky. The edges of the RAM chips containing high Au content were stamp-sheared from the RAM chips and are referred to as Au fingers. The compositions in original Au fingers are shown in the results section. 

Preliminary size reduction of cut Au fingers was achieved by a knife mill (Retch SM 300) to a top size of 3.4 mm. Further size reductions to top sizes of −2 and −1.2 mm were carried out using a laboratory analytical mill (Model: Analytical Mill 4301-00, manufactured by Cole-Parmer, Vernon Hills, IL, USA). The shredded Au fingers were then leached in Cu-ammoniacal solution as a pre-step to remove existing Cu in the RAM chips, prior to Au thiosulfate leaching. The Cu-ammoniacal solution was made from 1 M (NH_3_)_2_SO_4_, 4 M NH_4_OH, and 40 g/L Cu^2+^. The lixiviant was chosen based on the previous results on Cu leaching to maximize the copper removal. The residues after Cu pre-leach were then rinsed three times with deionized water and dried in a drying oven at 70 °C overnight. The decopperized Au fingers were fed to the subsequent Au thiosulfate leaching. The developed flowchart of feed preparation and leaching experiments is depicted in Figure 2. 

### 2.2. Leaching Experiment

Au thiosulfate leaching tests were carried out using 10 g/L pulp density at a constant stirring speed for 4 h; 10 g/L pulp density was chosen to produce approximately 100 ppm Au solution, if assuming 100% recovery. Leaching tests were conducted under alkaline conditions, which equated to about pH 9.6. During leaching, Eh, pH, and temperature were monitored by a multifunctional pH/ORP/ATC probe (Model: InPro 3100i, manufactured by Mettler-Toledo, Columbus, OH, USA). Leachate samples were taken at 0, 15, 30, 60, 120, and 240 min. 

To provide additional rationale for the selection of leaching parameters, the following explanation is provided. As depicted previously in Figure 1, the chemical reactions involved in Au dissolution in a Cu–ammoniacal thiosulfate system can be further categorized by the aerobic and anaerobic, depending on the oxidation mechanism.

In aerobic leaching, where oxygen is continuously supplied to oxidize Cu(I) (as Cu(S_2_O_3_)_3_^5−^) back to Cu(II) (as Cu(NH_3_)_4_^2+^), in the cathodic reaction, as shown in Equation (6), Cu(NH_3_)_4_^2+^ is regenerated and continually acts as the oxidizer to leach Au. Under regeneration by aeration, only a small amount of Cu(NH_3_)_4_^2+^ is consumed; thus, it is regarded as the catalyst [45]. However, in the presence of excessive oxidizer in the solution, thiosulfate decomposition is then enhanced to form the breakdown product, such as tetrathionate, as indicated by Equation (7). 

Cu(II) regeneration (aerobic leaching): (6)$$\mathrm{Cu}{\left(S_{2}O_{3}\right)}_{3}{}^{5-}+4{\mathrm{NH}}_{3}+\frac{1}{4}O_{2}+\frac{1}{2}H_{2}O\to \mathrm{Cu}{\left({\mathrm{NH}}_{3}\right)}_{4}{}^{2+}+3S_{2}O_{3}{}^{2-}+{\mathrm{OH}}^{-}$$

Cu(II)-promoting decomposition: (7)$$2\mathrm{Cu}{\left({\mathrm{NH}}_{3}\right)}_{4}{}^{2+}+8S_{2}O_{3}{}^{2-}\to 2\mathrm{Cu}{\left(S_{2}O_{3}\right)}_{3}{}^{5-}+8{\mathrm{NH}}_{3}+S_{4}O_{6}{}^{2-}$$

In anerobic leaching, Cu(NH_3_)_4_^2+^ acts as the predominant oxidizer to extract Au, being constantly consumed without further regeneration. In an oxygen-excluded system, the decomposition of thiosulfate by Cu(NH_3_)_4_^2+^ is limited (Equation (7)), which contributes to improving the leaching kinetics of Au [46], but to efficiently leach Au with Cu(II) as the oxidizer under an oxygen-eliminated condition, the concentration of Cu(II) needs to be maintained at a certain level to control the solution potential [45], while not being high enough to promote thiosulfate decomposition [46,47]. As such, a controlled aeration may be needed to partially regenerate the Cu(II) and to maintain the solution Eh at a suitable level. 

From the above discussion, the effects of decopperization, aeration, initial Cu(II) concentrations, ammonium thiosulfate and ammonium hydroxide concentrations, particle sizes, and temperatures were studied and summarized in Table 2. The lixiviant consisted of ammonium thiosulfate (NH_4_)_2_S_2_O_3_), ammonium hydroxide (NH_4_OH), and additional cupric sulfate pentahydrates (CuSO_4_·5H_2_O) as the catalyst. Based on the feed assay, 0.25–0.75 M of (NH_4_)_2_S_2_O_3_, 0.5–1.0 M of NH_4_OH, and 25–75 mM of initial Cu(II) were chosen to provide excessive reactants and catalysts in the solution. 

During leaching tests, continuous aeration was provided by air sparging (21% O_2_ and 78% N_2_) at controlled rates (0, 60, and 120 L/min). In 0 mL/min aeration tests, argon gas was purged into the reactor to expel the existing oxygen in the system. Particles were shredded to produce 3 product top sizes: 3.4, 2.0, and 1.2 mm. The temperatures selected were 25, 35, 45, and 55 °C. All chemicals used in this study were reagent grade. De-ionized water was used in the chemical makeup, sample preparation, and all the leaching experiments.

For convenience and clarity, the experimental settings are provided in the codex shown in Table 3. From these tests the following designs of experiments are proposed for statistical evaluation.

A three-level, two-factor design of experiments to evaluate the effects of copper concentration and aeration rate on leaching (comprising Experiment 2–10); Two experiments consisting of a three-level single-factor design testing changes in the concentrations of thiosulfate and ammonia, respectively (Part A, with changing thiosulfate Experiment 9, 13, 14; Part B, with changing ammonia Experiment 11, 9, 12; and Part C, which regresses 9, and 11–14);A three-level single-factor design to evaluate particle top size (comprising Experiment 15, 9, 16);A four-level single-factor design to evaluate the effects of temperature (comprising Experiment 9, and 17–19);A composite regression analysis including aeration, copper concentration, thiosulfate concentration, and ammonia concentration (comprising Experiment 2–4, 8–14). 

The discussion of the dependent variables and the statistical analysis will not be given at this time but is better suited for the discussion portion of the paper for reasons shown hereafter. 

### 2.3. Analytical and Characterization Methods

#### 2.3.1. Solid Assay

A chemical assay of solid samples, by roasting and acid digestion, was conducted to determine the concentration of metals contained in the solid phase. Representative solid samples were first pulverized by an analytical mill (Model: Analytical Mill 4301-00, manufactured by Cole-Parmer, Vernon Hills, IL, USA) to −30 mesh (600 µm), then roasted in the furnace at 580 °C for 8 h to remove the organic and volatile components. After roasting and homogenizing by pestle and mortar, representative 0.5 g of samples were taken for acid digestion. The weighed samples were placed in a 50 mL PFA (perfluoroalkoxy) digestion tube. Hydrofluoric acid (HF) and aqua regia (molar ratio of HCl:HNO_3_ = 3:1) were used as the digesting reagents. First, 20 mL of aqua regia and 20 mL of HF were added to dissolve metals and polymers. The digestion was conducted in a hot block at 145 °F (63 °C) for 5 h until the liquid evaporated completely. The final digested samples were further prepared for elemental analysis. 

#### 2.3.2. ICP-OES Analysis

ICP-OES (inductively coupled plasma-optical emission spectrometry), manufactured by Spectro Arcos, was used for elemental analysis. To eliminate the precipitation of sulfur species during sample preparation when transitioning from alkaline to acid matrix, the following sample preparation process was developed. Solution samples containing Au thiosulfate complex were oxidized by adding concentrated H_2_O_2_ to allow full oxidation of polythionate species to be stabilized as SO_4_^2−^. The oxidized samples were then acidified by adding aqua regia to stabilize Au^+^ by the complexation with Cl^−^. The prepared samples were further diluted in 5% HNO_3_ matrix for ICP-OES analysis. The metal recovery % is calculated by Equation (8): (8)$$Metal\;Recovery\;\left(\%\right)=\frac{{\left[Me\right]}_{solution}\times V_{solution}}{{\left[Me\right]}_{solution}\times V_{solution}+{\left[Me\right]}_{solid}\times m_{solid}}$$
where [*Me*]*_solution_* is the metal concentration in liquid phase (mg/L); *V_solution_* is the volume of liquid phase; [*Me*]*_solid_* is the metal concentration in solid phase (mg/kg); and *m_solid_* is the mass of solid phase. 

#### 2.3.3. SEM-EDS Characterization of Feed Materials

The untreated Au fingers, and the treated Au fingers after leaching experiments, with higher and lower Au recoveries, were characterized by a scanning electron microscope (SEM, Quanta 250, ThermoFisher Scientific formerly FEI, Hillsboro, OR, USA) equipped with energy-dispersive X-ray spectroscopy (EDS, X-Max detector, Oxford Instruments, Abingdon, UK). The detector used on SEM was an Everhart-Thornley detector (ETD) with adjusted bias to optimize signal intensity while minimizing charging effects in the image. Multiple fields of view in each sample were scanned automatically at high magnification using incident electrons with 30 keV energy and appropriate beam currents to balance signal intensity and spectrum accuracy. An accelerating voltage of 30 kV (at a working distance of 10 mm on the SEM) was used to excite X-rays from all potentially present elements and to maximize the amount of X-rays generated so as to accumulate numerous counts in each field of view. Automation of the acquisition over a large area, EDS spectra analysis, and maps generation were achieved using AZtec 6.0 software with TruMap algorithm.

## 3. Results and Discussion

The results experienced in this work were not anticipated and have not been observed in the literature regarding the thiosulfate leaching of e-waste where the Au recovery dropped precipitously after initial recovery. To explore the causes of this behavior, the decopperization results are provided, followed by the characterization of the RAM chips to explore the morphology of the leached substrate, which is followed by a discussion of cementation and then the specific leaching results. 

### 3.1. Effect of Decopperization

Because of the high solubility of Cu in thiosulfate solution, if it is not removed prior to Au leaching, can result in a high thiosulfate consumption, thus prohibiting the dissolution of Au (as described in Equation (7)). In addition, the ability of Cu to recover Au ions from thiosulfate solution, through a cementation reaction as discussed in the introduction, will lead to the deposition of leached Au thiosulfate [34]. This seems to be a contributing factor to the difference in recoveries reported by Ha et al. [39,42] and Tripathi et al. [40], as shown in Table 1. Therefore, it is important to remove as much Cu in the as-shredded Au fingers prior to Au–thiosulfate leaching. As described in the methods section, a decopperization pretreatment was carried out using a Cu-ammoniacal lixiviant containing 1 M (NH_3_)_2_SO_4_, 4 M NH_4_OH, and 40 g/L Cu^2+^, under air purging (to continuously regenerate the oxidizer) at 120 mL/min. A maximum of 90% Cu removal was achieved during decopperization, with a leaching time over 8 h, under sufficient oxidizer supply. The concentrations of the remaining compositions in Au fingers are given in both Table 4 and Figure 3. It was suspected that the residual (~10%) Cu was masked by the Au and Ni layer, therefore unable to be extracted unless the Au and Ni were removed. 

To examine the effect of decopperization as a feed preparation step, two Au leaching scoping tests were carried out using original Au fingers and decopperized Au fingers as feed materials. As indicated in Figure 3, it is clear that Au extraction % in thiosulfate solution was largely improved by decopperization. The Au recovery after decopperization was 88.7% compared to 12.7% when leaching for 4 h. It was observed that the extraction of Ni showed the inverse result to Au, with a recovery rate of 74.1% and 1.4%, in the original and decopperized Au fingers, respectively. It was hypothesized that, in untreated Au fingers, sufficient available Cu existed to influence the quantity of Au ions in solution. Further supporting evidence for this hypothesis will be discussed in the following sections. From these results, it is observed that Cu has a significant role in the recovery of Au and that a pretreatment step is critical to the effective recovery of Au. 

### 3.2. SEM Characterization of Untreated Au Fingers

The as-shredded Au fingers were analyzed by SEM-DES. Results of a single Au-finger flake, dissociated from board substrates and the Au fingers with a sheared surface are shown in Figure 4 and Figure 5, respectively. 

As indicated in Figure 4, the most abundant elements in the Au fingers were Au, Ni, and Cu, while Al was found in the board substrate. The EDS mapping of Au, Ni, and Cu indicates their association with each other in the electric contacts of the Au fingers. In the board substrates, Al was found to be the predominant element, as shown in Figure 4.

The alignment of Au, Ni, and Cu indicated the existence of a Au-Ni-Cu lamellar structure. Regardless, it was difficult to determine the order of the Au-Ni-Cu layers, i.e., the respective orientation of these elements from top to bottom by SEM, as shown in Figure 4. To be able to determine the exact order, a Au-finger flake presenting a sheared surface was selected, as shown in Figure 5. The orientation of the Au > Ni > Cu layers seems to be determinable from the shape and markings observed in the Ni plot coincidently matching the shape of scratches on the Au surface. In reviewing Figure 5, the Ni begins to be observed corresponding to areas where Au has begun to be removed. As such, Ni was likely the next layer below the Au surface layer. Moreover, Cu was only exposed at the sheared edge at the cross-section of the Au-finger flake, which seems to indicate shearing rather than surface abrasion and that the Cu layer is below the Ni and Au layer. Based on this information, it is strongly suspected that the order is Au > Ni > Cu. This is also corroborated by typical plating techniques using a Ni layer over Cu to enable plating of Au. This finding informed the substrate ordering shown in Figure 1.

### 3.3. Mechanistic Exploration of Time-Dependent Au Thiosulfate Leaching Results

#### 3.3.1. Investigation on the Aeration/Oxidation in Au Thiosulfate Leaching System

As summarized in Table 2, the leaching of Au in Cu-NH_3_-S_2_O_3_ solution using decopperized Au fingers was evaluated for the effects of Cu(II) concentration, aeration, thiosulfate and ammonia concentrations, size, and temperature. For the purpose of exploration, the Au-leaching mechanism in Cu-NH_3_-S_2_O_3_ solution under anaerobic condition (without aeration, under argon purging) is presented in Figure 6, with various Cu(II) concentrations to provide the initial oxidizer. As can be seen, the leaching of Au did not meet initial expectations nor any observed behavior in the literature, save for one reference of Au-ore leaching [48]. This behavior can be summarized in the observed Au recovery in Figure 6, which seems to show typical leaching recovery followed by Au loss in the lixiviant as a function of time. This behavior would seem to be described either by a precipitation reaction or by a cementation reaction causing the re-recovery of Au on an exposed Cu surface after the removal of Au and Ni. That this behavior is likely caused by a cementation reaction is supported by SME-EDS post-leaching characterization of the leached residues. To further demonstrate the observed repeatability of this phenomena, a replicate was performed and discussed more appropriately in a later section. 

SEM-EDS characterization was conducted on two leaching residues to explore the potential causes of the significant recovery differences corresponding to aerobic and anaerobic conditions. This corresponded to Experiment 9 with 88.7% Au recovery under aeration (Figure 7a) and Experiment 2 with only 1.19% Au recovery under no aeration (Figure 7b), where suspected cementation or precipitation of Au occurred. By comparing the EDS mapping of two residues (Figure 7a,b) side by side, it is clear to see that Au, Ni, and Cu were closely associated with each other in the two residues, further supporting the importance of the morphology of the overlaying Au-Ni-Cu layers. Notably, the mapping of sulfur showed two distinct patterns in Figure 7a,b. In Figure 7a, as Au leaching reached a steady state, S occurred in a more dispersed pattern, with little correlation with other elements and no distinguishable shape. However, in Figure 7b, the occurrence of S changed to a pattern closely associated with Au, Ni, and Cu. 

The contrast of S occurrence between these two leaching tests suggests the precipitation of a form of S species when Ni leaching occurred by allowing the exposure of Cu. However, it was difficult to identify the form of precipitated sulfide species. The average weight % of the elements concerned, i.e., Ni, Cu, Au, and S, are shown in Figure 8 from the EDS analysis. The leaching residues corresponding to Experiment 9 (88.7% Au recovery) (Figure 7a) exhibited a higher content of Ni remaining in the solid phase, which was consistent with the observation of low Ni recovery being requisite to high Au recovery. In contrast, the leaching residues of Experiment 2 with low Au recovery (1.4%) (Figure 7b) exhibited 42.2% Ni recovery, which corresponded to a lower observed weight % of Ni by EDS. These results of lower observed Ni in the solid phase agreed with the observations of increased Ni in the leaching solutions. Moreover, a non-negligible amount of Cu content (5.9% wt.) was detected with a 1.3% increase in S for Experiment 2, suggesting the EDS detection of exposed copper corresponding to Au/Cu/S co-occurrence.

These results further confirmed the hypothesis that under anerobic conditions, Au once leached was redeposited either by the precipitation or cementation of Au in the presence of Cu that includes sulfur. Additional evidence corroborating these results was reported by Jeffrey et al., who observed the formation of Au_2_S in the presence of Cu sulfides in Cu(II)-NH_3_-S_2_O_3_ solution [49]. That passivation occurred cannot be dismissed summarily, as Nie et al. identified the formation of Cu in the passivation layer on the surface of Au in a similar thiosulfate system [25]. However, the high recovery initially, shown in Figure 6, would suggest that passivation did not occur. 

#### 3.3.2. Investigation on Solution Eh and pH in Au Thiosulfate Leaching System

In an attempt to further explore this topic in the experimental context provided, the time-dependent recovery behavior of Au and Ni (a) and pH and Eh (b) were monitored and plotted as shown in Figure 9, corresponding to aeration experiments 3 and 9 in comparison to experiments 12 and 14, which varied the ammonia thiosulfate and ammonia hydroxide concentrations shown in Figure 10. In the leaching tests with (Experiment 9) and without aeration (Experiment 3), there was no noticeable difference in the pH, but the difference in the Eh was quite significant, as indicated by Figure 9b. It was observed that under 120 mL/min aeration (Figure 10), a higher Au recovery with Ni extraction was achieved. Correspondingly, solution Eh decreased first and then maintained at 180 mV (vs SHE). On the other hand, with no additional aeration (Figure 9), Au was leached first but decreased as Ni extraction was initiated. In this case, the corresponding Eh showed a similar decreasing trend in the first hour but kept declining to 40 mV (vs. SHE) because of the depletion of oxidizer Cu(II) and limitation of its regeneration. A hypothetical explanation is that, in the anerobic experiment, Ni was extracted, exposing the underlying Cu, leading to Au extraction from the lixiviant. Without an additional oxidative source, the Eh corresponding to the shift in the Cu(II)/Cu(I) ratio did not recover and, as Cu was oxidized to Cu(I), additional Cu(II) was not sufficiently available to re-leach the Au.

This is in contrast to the leaching tests shown in Figure 10 comparing the markedly different pH and Eh behaviors between Figure 9b and Figure 10b. Close examination of Figure 10b shows very little variation between the normal plot shown by Experiment 12 (higher pH caused by additional ammonia hydroxide) and that of Experiment 14 (higher ammonia thiosulfate and lower ammonia hydroxide). The Eh concluding around 0.15 V, as indicated in Figure 10b, suggests a similar Cu(II)/Cu(I) ratio caused by oxidation. Thus, it would appear that Au reduction is then temporary and localized, as indicated by the rebound in Au recover supposedly in conjunction with Cu removal. This hypothesis is also supported by observing the effects of Au recovery as Ni extraction approached equilibrium; the precipitated Au (on the Cu surface) was re-leached by the sufficient thiosulfate in the solution and the sufficient oxidizer supply in maintaining the favorable Eh. Careful comparison of Figure 10b may suggest slight and faster decreases of Eh linked to the solution potential favoring the re-deposition of Au, but that also may be within the realm of experimental error. Of further note are the similarities in the first datapoint of Figure 10a, which shows the independence of the thiosulfate concentration suggesting, at least in these experiments, mass transport kinetic dependence. Figure 9a is also very close kinetically. 

In comparing Figure 9 and Figure 10, it is revealed that it is critical to maintain the Eh above 0.15 V for the purpose of preventing Au precipitation under reductive potential. In Figure 9, as Cu(II) was consumed, the solution Eh was lowered to less than 0.15 V, and the precipitation of Au occurred, whereas in the presence of sufficient oxidizer, as indicated by Figure 10, the leaching of Au might be depressed temporarily, but it continued to proceed after 2 h. Similar results are reported by Feng et al. in an anaerobic system where the mixed solution potential is largely dependent on the redox couple of Cu(II)/Cu(I) [45]. A critical control in the anaerobic system is to maintain the mix potential above 0.15–0.19 V (vs. SHE) [45]. Consistently, the electrochemistry study by Nie et al. indicated that the reduction of the Cu(II) complex in the NH_3_-S_2_O_3_ system occurred at a potential of 0.15 V, and the black Cu sulfide was formed on the surface of Au after the reduction of the Cu(II) complex [25]. 

To assist in understanding the effects of Eh in this system, an Eh-pH diagram is provided to demonstrate the species stability in such a system under different pH and Eh ranges. The Eh-pH diagram, including the Cu-NH_3_-S_2_O_3_ species in the current leaching system, was constructed using the HSC 9.0 program, as shown in Figure 11. The corresponding concentrations of Cu, NH_3_, and S_2_O_3_ were 0.05 M, 1.5 M, and 1 M, respectively. As indicated by the Eh-pH diagram in Figure 11, at current leaching conditions, from pH 9 to 10, there were three possible Cu species across different Eh ranges from 0 to 0.25 V. Under reductive conditions (at a lower Eh range below 0 V), Cu stably exists at Cu_2_S, indicating the formation of Cu_2_S precipitates, which further results in the co-precipitation of Au ions [35]. This may also be suggested by Figure 7b. In alignment with the leaching result presented in Figure 9, at a lower Eh without aeration, the drop of Au recovery occurred because of the formation of Cu sulfides and the co-precipitation of Au on the accessible Cu surface. Under more oxidative conditions, at a higher Eh above 0.15 V, Cu is stabilized in the solution as Cu(NH_3_)_2_^+^ and Cu(NH_3_)_4_^2+^, which leads to the continuous extraction of Au and, eventually, a higher Au recovery. As the stability regions of Cu(NH_3_)_2_^+^ and Cu(NH_3_)_4_^2+^ species are greatly governed by the redox couple of Cu(II)/Cu(I), it was therefore critical to maintain the solution Eh by controlled aeration. Evidenced by the result shown in Figure 9, at a maintained Eh above 0.15 V with sufficient oxidizer (Cu(NH_3_)_4_^2+^) and complexing ions (NH_3_ and S_2_O_3_^2−^), although a drop of Au recovery was intended at the beginning, the re-dissolution of Au was initiated after Ni extraction started to reach its equilibrium. 

From the above observations, it appears that Ni extraction was either coincident to or casual of the decrease in Au recovery, with the reasons behind this phenomenon remaining unclear. In considering the mechanism of decreasing Au recovery shown in Figure 6, Figure 9 and Figure 10, it appears that deposition of Au may have occurred to remove Au from the leachate solution. It is hypothesized that this can occur in a number of ways. The first is thiosulfate consumption by the extraction of Ni. As Ni was extracted, it competed with Au on complexing the thiosulfate ions and caused the leached Au+ to lose its complexation with thiosulfate. However, as calculated, the thiosulfate concentration was sufficient even though all the Ni and Au was extracted in solution. It is less likely that the enriched Ni concentration would result in the full depletion of thiosulfate ions during Au leaching. 

The second possibility is the cementation of leached Au on the newly exposed Cu surface. The identification of Au-Ni-Cu interlayers was previously revealed by SEM-EDS (as shown in Figure 5), and the high possibility of Au re-deposit on Cu during thiosulfate leaching was hypothesized accordingly. As Au was successfully leached, the Ni underlayer was exposed and reacted with thiosulfate lixiviant. As Ni was leached, a Cu layer was exposed, which led to the cementation of Au on Cu. Although over 90% of the Cu in the waste chips was removed by decopperization pretreatment, the extraction of Ni exposed a residual Cu surface beneath it. As a result, the leached Au thiosulfate ions were deposited on the newly exposed Cu surface, via a cementation reaction. 

Jeon et al. studied the electrochemistry of Au(S_2_O_3_)_2_^3−^ deposition on the accessible Cu surface in a Au thiosulfate system, as expressed in Equation (9) [34].
(9)$$\mathrm{Au}{\left(S_{2}O_{3}\right)}_{2}{}^{3-}+e^{-}\to \mathrm{Au}\;+2S_{2}O_{3}{}^{2-}$$

They discovered that Au deposition in ammoniacal thiosulfate solution was enhanced by increasing Cu concentration up to 50 mM. Furthermore, their SEM-EDS results showed that the deposition of Au occurred together with the deposition of Cu [35]. Moreover, in a heterogeneous material like PCBs, the presence of Al tends to stimulate Au deposition on Cu [34]. This could explain the leaching trends of Ni and Au, where Au extraction was increased at the beginning but dramatically dropped down as soon as Ni was extracted. In our study, the deposition of Au with Cu and S was confirmed by the SEM-EDS characterization of post-leaching residues, which is discussed in a later session. 

Combining with the previous leaching results and the corresponding Eh-pH diagram (as shown from Figure 9, Figure 10 and Figure 11), under reductive leaching, Cu sulfides become the dominant species stabilized in the system. As further proved by the SEM-EDS, under an unfavored leaching condition, with solution Eh below 0.18 V, thiosulfate degradation, together with the formation of Cu sulfides, occurred simultaneously, which caused Au cementation on the available Cu surface. Therefore, the decrease in Au recovery was observed during leaching as a response to the Ni extraction. From a processing point of view, there were a few recommended conditions that could benefit the redissolution of Au after its deposition, given sufficient aeration and Cu(II) supply. These conditions include a higher NH_3_ concentration (or lower AT/AH ratio), excessive thiosulfate ions, or mildly elevated temperature. 

### 3.4. Presentation and Analysis of Leaching Results

With the discussion of the observed loss of leached Au from the lixiviant, the leaching experiments may now be discussed. Initially, it was anticipated that a typical kinetic model would be developed from the leaching data, but as discussed previously, the leaching kinetics are confounded by a secondary response, which is the cementation or precipitation upon the leaching of Ni. For this reason, it is proposed that three responses may be valuable in interpreting this system. The first is recovery vs. time plots for Au and Ni to show the interplay between leaching and precipitation/cementation. The second response proposed is a time to Au recovery drop. This is the latest time when Au recovery is observed to behave normally before a Au decrease. As shown in Table 5, not all recoveries are the same when this behavior is observed, so a third response is proposed called leaching proclivity. The leaching proclivity is calculated according to Equation (10). This is a composite response based on the geometric mean time of when the Au drop occurs and the recovery at the point of Au drop. The time is factored in by using the observed geometric mean time of Au drop divided by the full time of leaching (240 min). Thus was the Au drop observed at a time of 120 min, and the time weigh factor would be 2.0. The justification for this is a half decrease in time to leach that would result in a two-times increase in process throughput. The summary results are shown in Table 5. In this manner, normal statistical analysis may be conducted. The details of the results and analysis will be elucidated by experiment in subsequent sections.
(10)$$\mathrm{Leaching}\;\mathrm{Proclivity}=\frac{\mathrm{geometric}\;\mathrm{mean}\;\mathrm{time}\;\mathrm{of}\;\mathrm{observed}\;\mathrm{Au}\;\mathrm{drop}}{240\min\;\left(\mathrm{full}\;\mathrm{length}\;\mathrm{time}\right)}\times \mathrm{Au}\;\mathrm{recovery}\;\mathrm{at}\;\mathrm{time}\;\mathrm{of}\;\mathrm{drop}.$$

#### 3.4.1. Effect of Cu(II) Concentration and Aeration

Using the decopperized Au fingers as feed, the effects of the oxidizers, Cu(II) and O_2_, were investigated by varying the Cu(II) concentrations and the aeration rates. Based on the stoichiometry of Au leaching reactions, the minimum amount of Cu(II) needed was 12.71 mM, according to the feed assay. In order to provide sufficient Cu(II) as an oxidant, but not overdosing and creating an extremely oxidative leaching condition, the initial Cu(II) concentrations were chosen as 25, 50, and 75 mM. To maintain the Cu(II) regeneration and catalyzation during leaching, the aeration rates were chosen as 0 (argon purging), 60, and 120 mL/min. The result of varying Cu(II) concentrations (25, 50, and 75 mM) under a fixed aeration rate (0, 60, and 120 mL/min) are shown in Figure 6, Figure 12 and Figure 13. 

Referring back to Figure 6, Au recoveries increased at the beginning of leaching and then decreased dramatically once Ni recoveries commenced. This phenomenon occurred in all three tests using different Cu(II) concentrations with no aeration. However, as the initial Cu(II) concentration increased from 25 to 75 mM, the cross point of Ni/Au was postponed from 120 to 240 min. Without the additional aeration, the initial input of Cu(II) acted dominantly to oxidize Au and Ni, while the regeneration of Cu(I) to Cu(II) was limited by the lack of additional oxygen supply. 

As the aeration increased to 60 mL/min (Figure 12), leached Au thiosulfate complex was successfully stabilized in solution for 4 h, using 50 and 75 mM Cu(II) as initial oxidant. As indicated in Figure 12, 84.5% and 80.3% of Au recoveries were achieved using 50 and 75 mM Cu(II), respectively. Meanwhile, there was little Ni extraction, under 50 and 75 mM initial Cu(II). However, 25 mM of Cu(II) was not sufficient to maintain the leached Au under 60 mL/min aeration. Similar to the previous result, at Cu(II) 25 mM, 60 mL/min aeration, Au recovery reached its optimum in the first hour but then dropped down to 14.8% once Ni was leached in solution. The results of Au and Ni extractions using 120 mL/min of aeration were shown in Figure 13. The observed leaching trends and the interaction of Au and Ni extraction were similar to those using 60 mL/min aeration. Au recovery was slightly improved to 88.7% and stabilized during a 4-hour leaching test, using 50 mM Cu(II) and 120 mL/min aeration. 

The effect of the aeration rate on the leaching proclivity was studied and analyzed using the factorial regression analysis tool of Minitab. The first analysis was 60 mL/min to 120 mL/min to show the statistical insignificance between 60 mL/min and 120 mL/min, suggesting that the effects of the 60 to 120 mL/min are nearly identical. The nearness of the results can be seen in comparing Experiment 5–7 to those of 8–10 in Table 6. The results of the two-level two-factor analysis shows that, save for Cu(II) concentration, there is no statistical difference between the two levels. 

From this analysis, the 0 mL/min to 120 mL/min aeration rates were analyzed and presented in Table 7, which showed that the aeration rate and copper concentration become the significant factors. When 120 mL/min are compared to the argon (anerobic) experiments, it quickly becomes apparent that both aeration and Cu(II) are significant with a high level of interaction. The derived model from this analysis is shown in Equation (11). The conclusion from these analyses is that 60 mL/min is above the needed threshold for the effective aeration rate for the oxidation of Cu in solution. The associated model is given in Equation (11) as follows.
(11)$$\mathrm{Leaching}\;\mathrm{Proclivity}\;=-69.05\times \;c\left[{\mathrm{Cu}}^{2+}\right]+35.45\times \mathrm{Aeration}+32.22\times c\left[{\mathrm{Cu}}^{2+}\right]\times \mathrm{Aeration}+142.09$$
where $c\left[{\mathrm{Cu}}^{2+}\right]$ is the Cu concentration in units of millimolar (mM), and Aeration is the aeration rate in units of mL/min. 

#### 3.4.2. Effect of Ammonium Thiosulfate to Ammonium Hydroxide Concentrations (AT/AH Ratios and Concentrations)

As the deposition of Au was observed when using limited Cu(II) and aeration, this section investigates the effect of 50 mM Cu(II) concentration and 120 mL/min of aeration whilst varying ammonium thiosulfate and ammonium hydroxide. Two sets of experiments were carried out to study the effect of the thiosulfate-to-ammonia ratio (AT/AH ratio). In the first set, NH_3_ concentrations varied from 0.5, 0.75, and 1.0 M, while the S_2_O_3_ concentration was held constant at 0.5 M; this corresponded to AT/AH of 1, 0.67, and 0.5. The results of varying NH_3_ concentrations are shown in Figure 14. In the second experimental set, S_2_O_3_ concentrations varied from 0.25, 0.5, and 0.75 M, whilst NH_3_ concentration was kept at 0.5 M. The AT/AH ratio in the second set corresponded to 0.5, 1, and 1.5. The results of varying the S_2_O_3_ concentrations are shown in Figure 15. To validate the repeatability of the observed bimodal response, namely leaching-cementation behavior, a replicate experiment using the exact same condition was performed and is shown in Figure 16. 

As indicated in Figure 14, increasing the NH_3_ concentration corresponded to an increase in the rate of initial Au recovery. At lower NH_3_ concentrations of 0.5 and 0.75 M, Au leaching continued through the duration of experiment, as opposed to at the higher NH_3_ concentration of 1.0 M (AT/AH ratio of 0.5 M/1.0 M), which corresponded to a slight decrease in Au recovery at 180 min, concurring coincidentally with an increase in Ni extraction. Nevertheless, under higher NH_3_ concentration, the detrimental impact of Ni co-extraction on Au recovery was not as significant, as previously disclosed (as indicated in Figure 6, Figure 12 and Figure 13). 

It has been discussed in the literature that it is important to maintain the NH_3_ level in a thiosulfate solution, for the following reasons: (1) to provide sufficient NH_3_ complexing ligand to stabilize Cu(NH_3_)_4_^2+^ as oxidizer and the intermediate Au(NH_3_)_2_^+^ specie [46]; (2) to condition the solution pH to maintain the Au thiosulfate species at a stability region [50]; and (3) to prevent the formation of sulfurs and sulfides on the surface of Au and prevent Au deposition on Cu by dissolving the exposed Cu surface [28]. Accordingly, as the excessively available NH_3_ was preferentially adsorbed on the surface of Cu, the deposition of Au on Cu was prevented. 

In Figure 15, when the AT/AH ratio increases to 1.5 (AT/AH of 0.75 M/0.5 M), it appears the leaching of Ni was possible, allowing for the same observed phenomena of a corresponding decrease in Au concentration. It was suspected that as Au and Ni were leached, the Cu layer beneath was released, which resulted in the cementation of Au on Cu. Thus, a decrease in Au recovery corresponding to the increase in Ni recovery occurred. Interestingly, with an excessive amount of thiosulfate, given enough residence time, Au was re-dissolved after 2 h, while Ni extraction started to reach its equilibrium. 

It was suggested by these results that a slightly higher concentration of NH_3_ than S_2_O_3_ (i.e., AT/AH ratio from 0.5 to 1) facilitated the Au extraction in two ways. First, the excessive amount of NH_3_ provided sufficient complexing ions to stabilize the leached Au ions (as Au(NH_3_)_2_^+^), thus minimizing the competing impact by Ni extraction. Second, a higher amount of NH_3_ prevented the deposition of Au on Cu. As reported by other researchers, NH_3_ not only acts as a complexing ligand to stabilize Cu and Au species in thiosulfate solution, Cu(NH_3_)_4_^2+^, as oxidizer and the intermediate Au(NH_3_)_2_^+^ specie [46] but also acts as a leaching agent to dissolve the exposed Cu surface below the Au layer, therefore preventing the leached Au from re-depositing on the Cu surface [28]. To understand the validity of the results in view of a potentially heterogeneous system, leaching was repeated under identical conditions, yielding largely similar results. 

With regard to statistical analysis, these two experiments comprised a three-level single-factor design varying the concentrations of thiosulfate and ammonia, respectively. The regression analysis for the change in ammonium hydroxide concentration, when the thiosulfate concentration is constant, showed the linear fit line for the given data and set of experiments, but the *p*-value of 0.121, which corresponds to Figure 17, showed some significance with regard to leaching proclivity when compared and studied against the constant thiosulfate concentration. Interestingly, the effect of the concentration of thiosulfate on leaching proclivity also becomes insignificant, based on *p*-Values of 0.332, when the concentration of ammonia remains constant and the thiosulfate concentration changes. As a final analysis of the regressed combined dataset, the resultant *p*-Values of 0.156 and 0.194 for thiosulfate and ammonia concentration, respectively, were achieved. Detailed results are available in Appendix A. 

#### 3.4.3. Effect of Particle Size and Temperature

The effects of particle size and temperature were studied, and the results are shown in Figure 18 and Figure 19. As indicated by Figure 18, by varying the top sizes of 3.4, 2.0, and 1.2 mm, larger particle sizes (3.4 and 2 mm) seemed to yield higher Au recoveries of 86.9% and 88.7%, respectively. Using the waste chips tested in this study, since most of the Au exists on the surface, the decrease in particle size from −3.4 to −2 mm had no significant impact on Au recovery. However, using a finer particle size of 1.2 mm, only 18.2% of the Au was leached after 4 h, while Ni recovery was improved to 50.8%. Similar observation on the opposite results between Ni and Au recoveries occurred when a new Cu surface was exposed during leaching (as indicated by the Au-Ni-Cu interlayer shown in Figure 5). As the particle size further decreased to −1.2 mm, kinetic Au leaching was faster; thus, the Ni underlayer was more readily exposed and rapidly leached. Then, the faster kinetics of the Ni removal led to the better accessibility of the underneath Cu layer, causing the Au re-deposition on the Cu surface in a shorter period of time. 

Similar observations indicating that a larger particle size could be more beneficial to Au leaching from PCB materials were reported by Ha et al. [39,42]. Using unshredded mobile phone scrap, they were able to achieve 90% Au recovery, without encountering the negative impact from Cu and Ni on Au leaching. Similarly, without the chemical pretreatment, Tripathi et al. reported a higher Au extraction of 78.8% using unshredded WPCBs, whereas the Au extraction was only 56.7% when using shredded WPCBs [40]. This is mainly because these metals were embedded in the laminates and unliberated from the PCBs matrix at a larger size. As a result, the thiosulfate attacked most of the Au mainly from the surface, while leaving the Ni and Cu in the board substrates. 

The results of Au thiosulfate leaching under various temperatures are shown in Figure 19. As thiosulfate decomposes at a higher temperature, the highest temperature studied in this work was 55 °C. By increasing the temperature from 25 to 55 °C, Au recovery decreased gradually, while Ni recovery was significantly enhanced, especially above 35 °C. As indicated by the ordinal table, the interruption time of Ni to Au recovery has been brought forward greatly at higher temperatures. For example, at temperatures above 35 °C, the dissolution of Au happened instantaneously and the Ni underlayer was leached within 15 min; therefore, the deposition of Au on Cu also occurred within a very short time at the beginning of leaching. After Ni extraction reached its equilibrium, from 120 to 180 min of leaching time, redissolution of Au took place, under the assistance of a sufficient supply of oxidizers (i.e., Cu(II) and oxygen) and complexation ions (i.e., S_2_O_3_ and NH_3_). 

#### 3.4.4. Combined Regression Model Excluding Particle Size, Temperature, and 60 mL/min Aeration

In the presence of factors including copper concentration, gas flow, ammonia thiosulfate concentration, and ammonia hydroxide concentration, factorial regression was performed. It showed that the ammonia thiosulfate is the only significant factor to a 0.05 confidence level contributing toward leaching proclivity, and the model for this scenario is presented in Equation (12). The results from analysis with *p*-Values are presented in Table 8, and the Pareto for these results is presented in Figure 20. Please refer to Appendix A for a detailed analysis table.
(12)$$\begin{array}{ll} \mathrm{Leaching}\;\mathrm{Proclivity} & =-69.1\times \;c\left[{\mathrm{Cu}}^{2+}\right]-12.7\times \mathrm{Air}\;\mathrm{Gas}\;\mathrm{Flow}+195.0\times c\left[\mathrm{AT}\right]-25.2\times c\left[\mathrm{AH}\right]+33.2\times \;c\left[{\mathrm{Cu}}^{2+}\right]\times \mathrm{Air}\;\mathrm{Gas}\;\mathrm{Flow}\\ & +139.6 \end{array}$$
where $c\left[{\mathrm{Cu}}^{2+}\right]$ is the Cu concentration in units of millimolar (mM), c[AT] is the ammonium thiosulfate concentration in units of molar (M), c[AH] is the ammonium hydroxide concentration in units of molar (M), and Air Gas Flow is the aeration rate in units of mL/min. 

### 3.5. Concerns and Potential Issues

In reviewing this work, there are a number of items that should be discussed and addressed, the first being the nature of the experimental design. In reviewing the experimental design, it can be surmised that the experimentation is largely scoping in nature. In this, only aeration was considered as being suitable for a factorial analysis. Other factors exhibit a single-factor analysis. In the context of the bimodal leaching response, this may be excusable because statistical analysis of a bimodal response is difficult and invalidates many typical statistical methods. 

A second point is that care must be exercised in interpreting the statistical analysis of the composite response of leaching proclivity. For example, Experiment 2, 5, 13, and 14 all exhibited nearly full Au recovery. Yet, when compared against leaching proclivity, the ranking is significantly different than the recovery. This was further confirmed by the R-sq value of 68.10% from the model presented in Section 3.4.4. This suggests that the given model does fully capture the variability exhibited by the leaching proclivity. Thus, care must be used to prevent misinterpretation of the analysis results. Future designs of experiments should utilize these learnings to eliminate the bimodal results exhibited here and utilize a response surface methodology. 

A third item of concern is the bimodal response of typical leaching followed by drops in recoveries. Upon review of the time-dependent recovery figures, it appears that there as some settings where Au recovery does not appear likely to improve again, versus those where it appears that Au recovery continues to improve after additional Cu leaching. In this, the leaching tests suggest that a longer leach time is warranted. With a suitable design that would eliminate the bimodal response, experiments using a response surface or central composite may be attempted. 

For future work, it is suggested that additional efforts should be devoted to characterization techniques to identify more completely the morphology of the sulfur/sulfide formation during the loss of Au from the lixiviant. The interaction of preferential leaching of Ni under certain conditions may also be worthy of additional investigation.

## 4. Conclusions

From the results of this study, the leaching of Au fingers from RAM chips exhibited what appears to be a bimodal leaching result, where first Au is recovered and then subsequently lost from the lixiviant. It is hypothesized that the existence of a Au-Ni-Cu lamellar structure was a significant contributing factor. This is based upon the observation that the co-extraction of Ni resulted in a conflicting effect on Au recovery because of the deposition of Au on the newly exposed Cu surface after Ni extraction. Supporting this statement, SEM-EDS characterization on the residues showed that the formation of reduced sulfur/sulfide species occurred under unfavored leaching conditions, which could further result in the co-deposition of Au. Furthermore, decopperization pretreatment was found necessary to remove the pre-existing Cu and greatly improve the Au recovery.

With regard to the analysis of data, due to the bimodal leaching response, a composite factor called leaching proclivity was developed to assist in the interpretation of the results. When aeration and copper concentration were evaluated concurrently, aeration was found important in facilitating the initial Au recovery by maintaining the Cu(II)/Cu(I) oxidation-reduction potential (ORP) in the leaching solution. 

The results further suggest that the decrease in particle size increased the possibility of Cu surface exposure after Ni extraction, thus accelerating the Au deposition. The elevation of temperature favored the Ni extraction but lowered the Au extraction. However, it appears that leaching in these conditions is somewhat incomplete owing to the increasing recovery with time. 

Last, this work presents interesting, potentially valuable insights into morphological reasons for the poor recovery of Au from e-waste. The interaction of Cu and the need for removal prior to and during Au recovery has been evidenced. 

## Figures and Tables

**Figure 1 materials-16-04940-f001:**
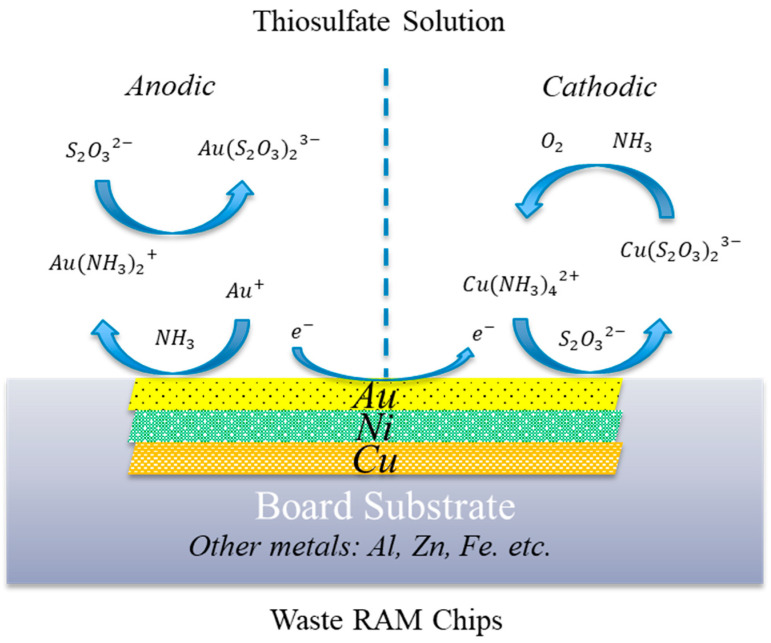
Schematical extrapolation of Au leaching mechanism in a Cu(II)-NH_3_-catalyzed thiosulfate system. Adapted from Xu et al. [26].

**Figure 2 materials-16-04940-f002:**
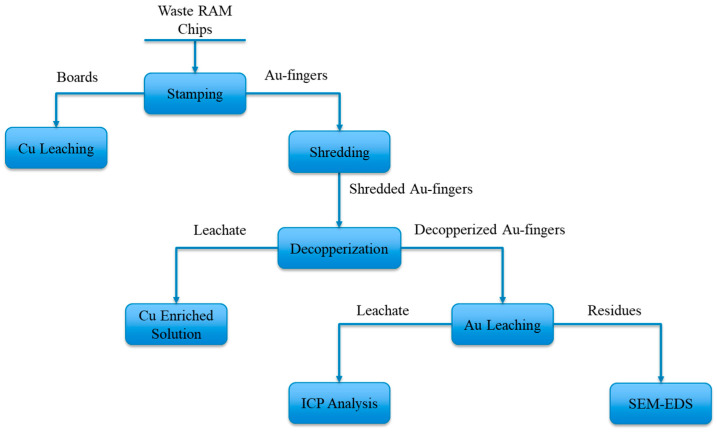
Flowchart of feed preparation, leaching process, and sample characterization in this study.

**Figure 3 materials-16-04940-f003:**
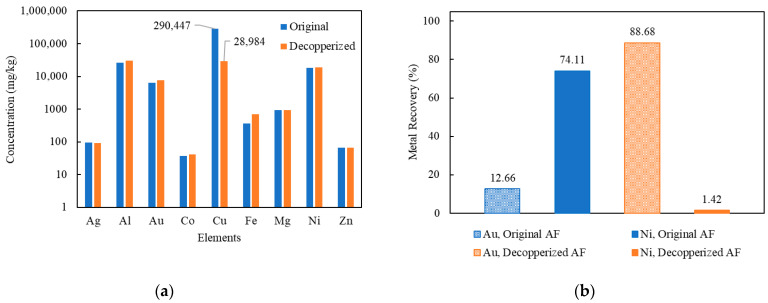
(**a**) Elemental concentration (mg/kg) in Au fingers before and after decopperization; (**b**) Au and Ni recovery (%) using original Au fingers and decopperized Au fingers (Experiment 1, 9; Cu(II) 50 mM; S_2_O_3_: 0.5 M; NH_3_: 0.5 M; aeration: 120 mL/min; Temperature: 25 °C).

**Figure 4 materials-16-04940-f004:**
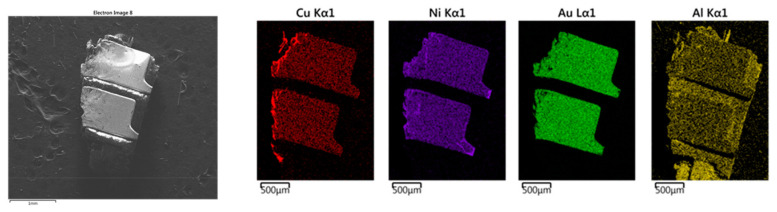
SEM images and elemental mapping of as-shredded RAM chips (Au fingers only).

**Figure 5 materials-16-04940-f005:**
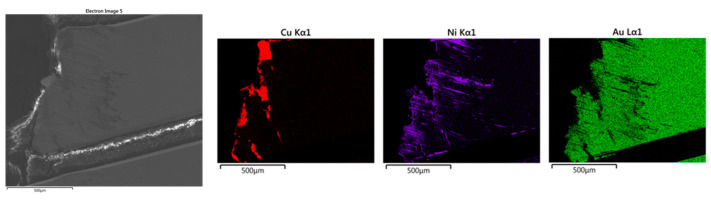
SEM images and elemental mapping of as-shredded RAM chips (Au fingers with the scratched surface).

**Figure 6 materials-16-04940-f006:**
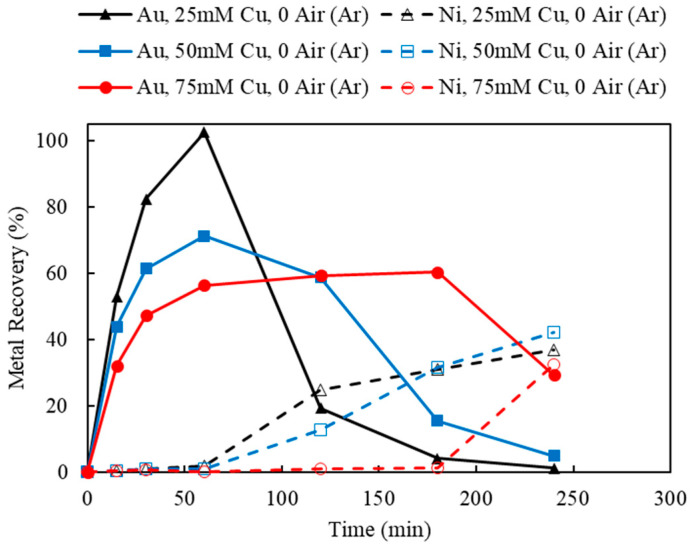
Au and Ni extraction (%) using 25, 50, and 75 mM Cu(II), under 0 mL/min aeration (argon purging) corresponding to experiments 2–4 in Table 3 (Experiment 2–4; S/L ratio: 10 g/L; particle size: −2 mm; S_2_O_3_: 0.5 M; NH_3_: 0.5 M; Temperature: 25 °C).

**Figure 7 materials-16-04940-f007:**
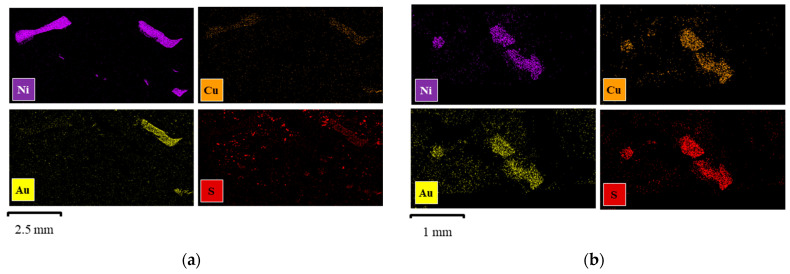
SEM images and elemental mapping of Au leaching residues: (**a**) high %R of Au under 120 mL/min aeration (Experiment 9) and (**b**) low %R of Au under Ar purging (Experiment 2).

**Figure 8 materials-16-04940-f008:**
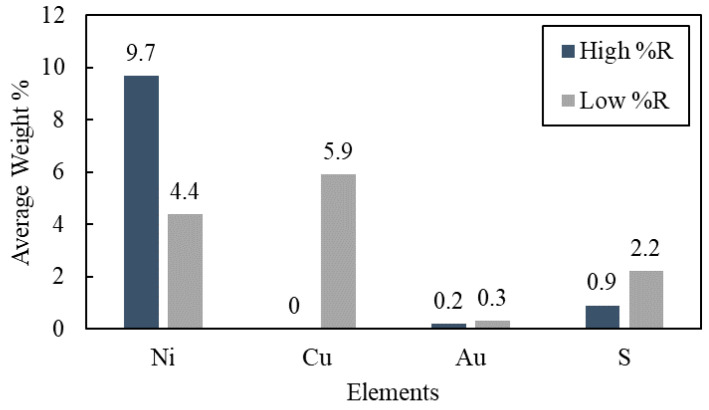
EDS average weight % of leaching residues (Experiment 9: high Au %R and Experiment 2: low Au %R, data corresponding to Experiment 9 and 2, using 50 mM Cu, AT/AH 0.5 M/0.5 M, under 120 and 0 mL/min aeration rates, respectively).

**Figure 9 materials-16-04940-f009:**
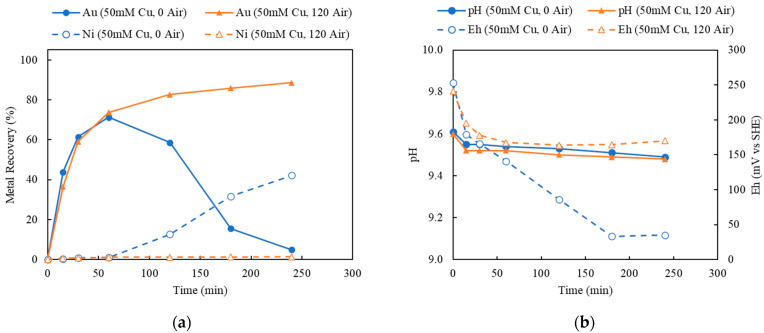
(**a**) Au and Ni extraction (%) and (**b**) pH and Eh (mV vs. SHE) in leaching tests under 120 mL/min air purging and Ar purging (Experiment 9, 3; S/L ratio: 10 g/L; particle size: −2 mm; Cu(II): 50 mM; S_2_O_3_: 0.5 M; NH_3_: 0.5 M; Temperature: 25 °C).

**Figure 10 materials-16-04940-f010:**
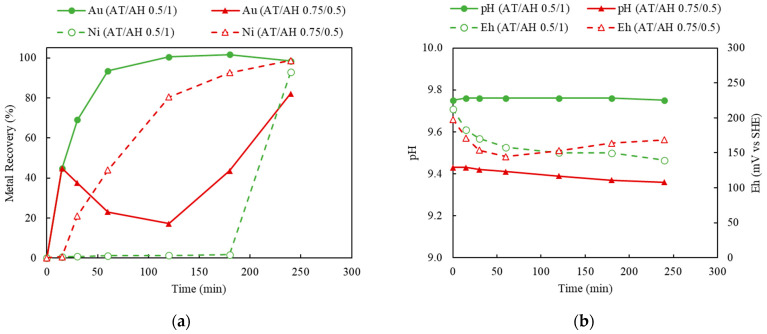
(**a**) Au and Ni extraction (%) and (**b**) pH and Eh (mV vs. SHE) in leaching tests with different AT/AH ratios of 0.5/1 and 0.75/0.5 (Experiment 14, 12; S/L ratio: 10 g/L; particle size: −2 mm; Cu(II) 50 mM; aeration: 120 mL/min; Temperature: 25 °C).

**Figure 11 materials-16-04940-f011:**
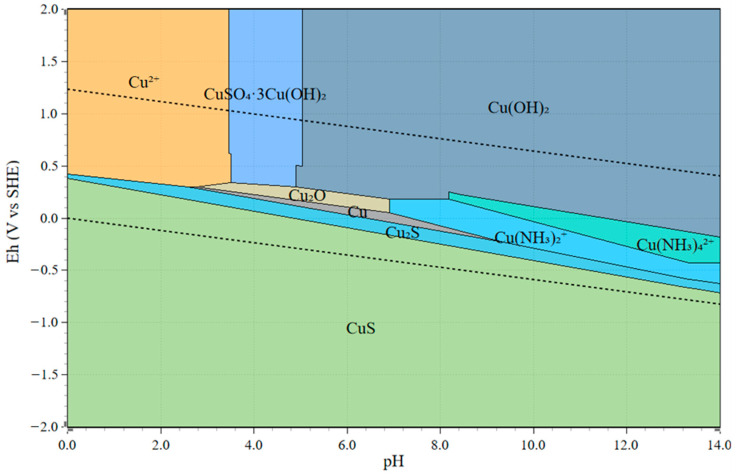
Eh-pH diagram of Cu-NH_3_-S_2_O_3_ system (Cu: 0.05 M; NH_3_: 1.5 M; S_2_ O_3_: 1 M, constructed by HSC 9.0).

**Figure 12 materials-16-04940-f012:**
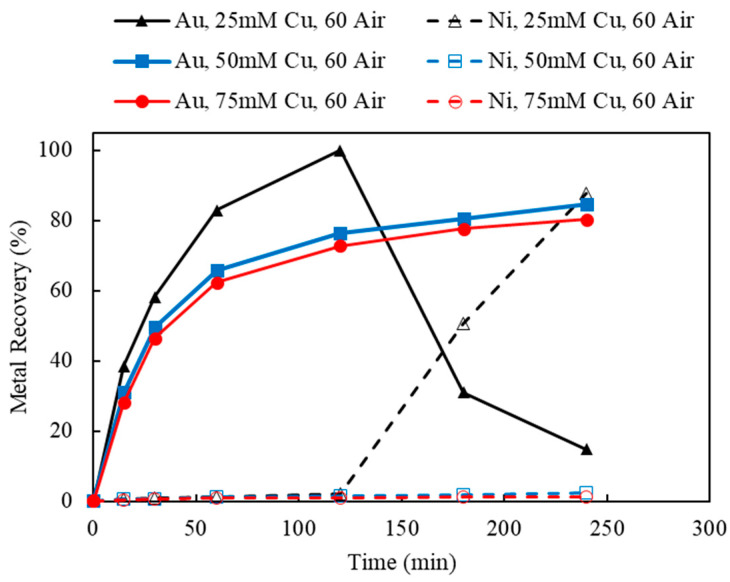
Au and Ni extraction (%) using 25, 50, 75 mM Cu(II), under 60 mL/min aeration (air purging) (Experiment 5–7; S/L ratio: 10 g/L; particle size: −2 mm; S_2_O_3_: 0.5 M; NH_3_: 0.5 M; Temperature: 25 °C).

**Figure 13 materials-16-04940-f013:**
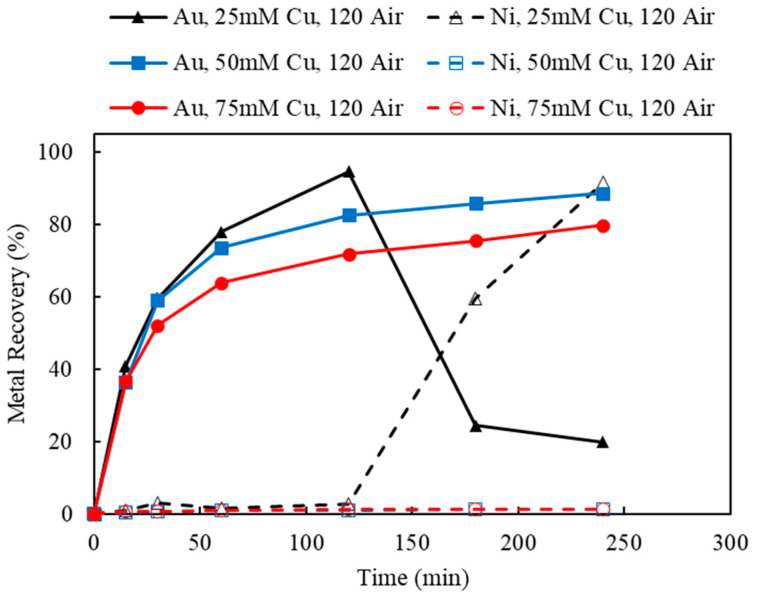
Au and Ni extraction (%) using 25, 50, 75 mM Cu(II), under 120 mL/min aeration (air purging) (Experiment 8–10; S/L ratio: 10 g/L; particle size: −2 mm; S_2_O_3_: 0.5 M; NH_3_: 0.5 M; Temperature: 25 °C).

**Figure 14 materials-16-04940-f014:**
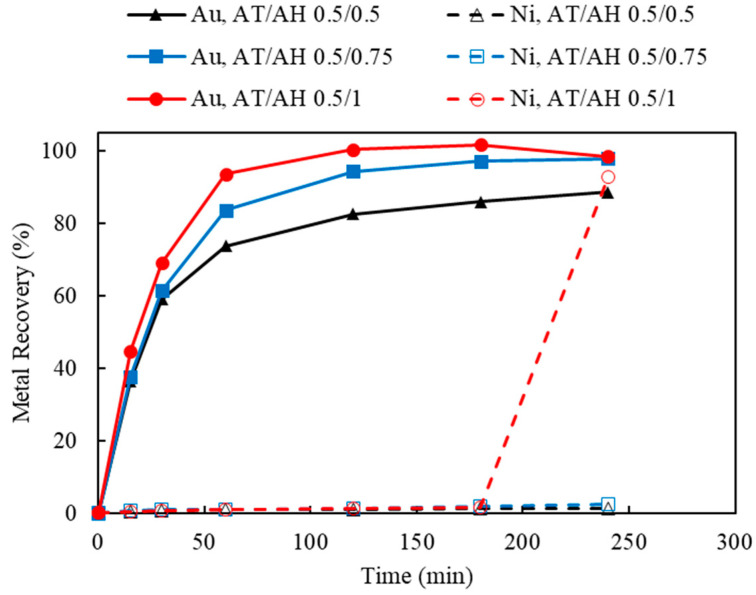
Au and Ni extraction (%) using AT/AH ratios of 0.5/0.5, 0.5/0.75, 0.5/1.0 (with fixed thiosulfate concentration of 0.5 M) (Experiment 9 and 13–14, S/L ratio: 10 g/L; particle size: −2 mm; Cu(II) 50 mM; aeration: 120 mL/min; Temperature: 25 °C).

**Figure 15 materials-16-04940-f015:**
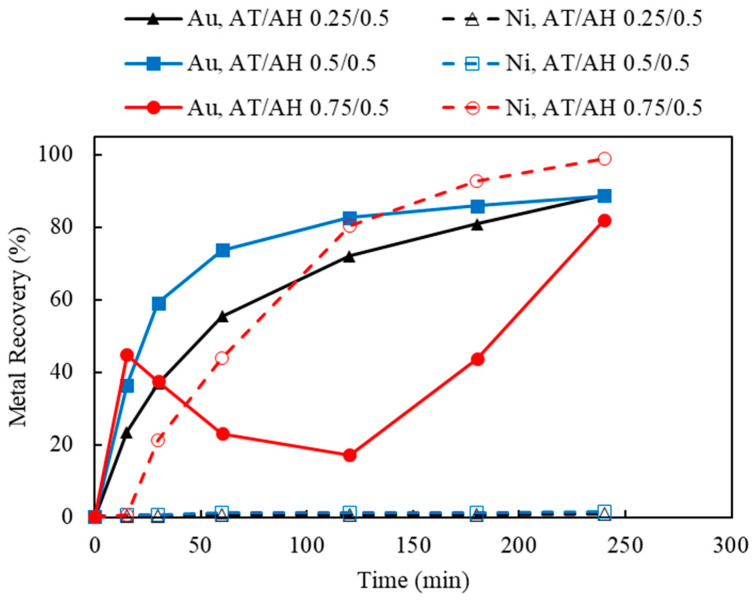
Au and Ni extraction (%) using AT/AH ratios of 0.25/0.5, 0.5/0.5, 0.75/0.5 (with fixed ammonia concentration of 0.5 M) (Experiment 9 and 11–12, S/L ratio: 10 g/L; particle size: −2 mm; Cu(II) 50 mM; aeration: 120 mL/min; Temperature: 25 °C).

**Figure 16 materials-16-04940-f016:**
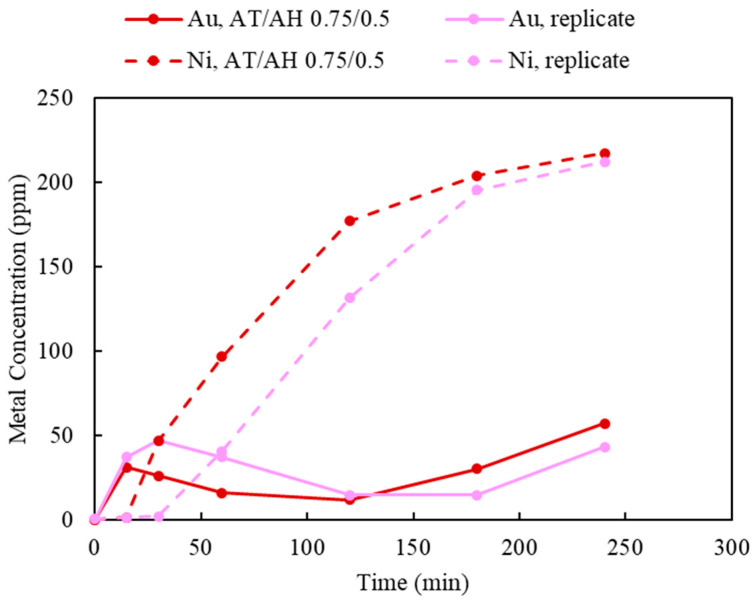
Replicate of leaching experiment using AT/AH ratio of 0.5/1.0 (Experiment 14 and its replicate; S/L ratio: 10 g/L; particle size: −2 mm; Cu(II) 50 mM; aeration: 120 mL/min; Temperature: 25 °C).

**Figure 17 materials-16-04940-f017:**
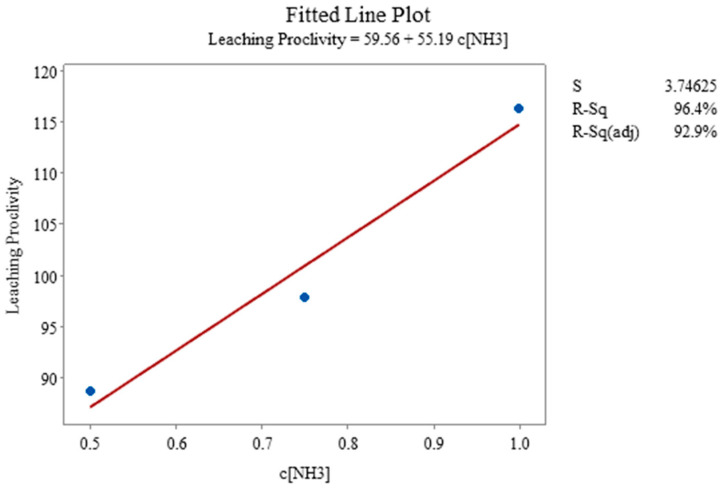
Fitted line plot for constant thiosulfate concentration when ammonia concentration was changing (Experiment 9, 13, 14).

**Figure 18 materials-16-04940-f018:**
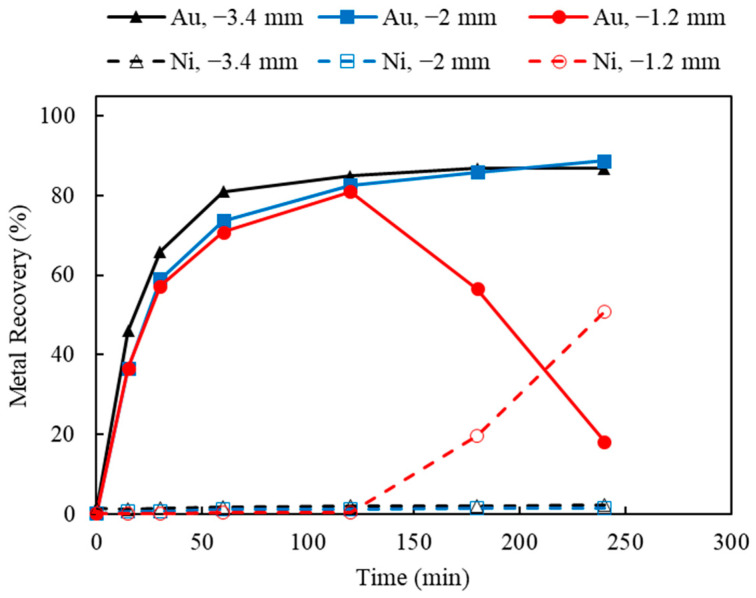
Au and Ni extraction (%) using AT/AH ratios of 0.5/0.5, 0.5/0.75, 0.5/1.0 (with fixed thiosulfate concentration of 0.5 M) (Experiment 9 and 15–16; S/L ratio: 10 g/L; particle size: −2 mm; Cu(II) 50 mM; aeration: 120 mL/min; Temperature: 25 °C).

**Figure 19 materials-16-04940-f019:**
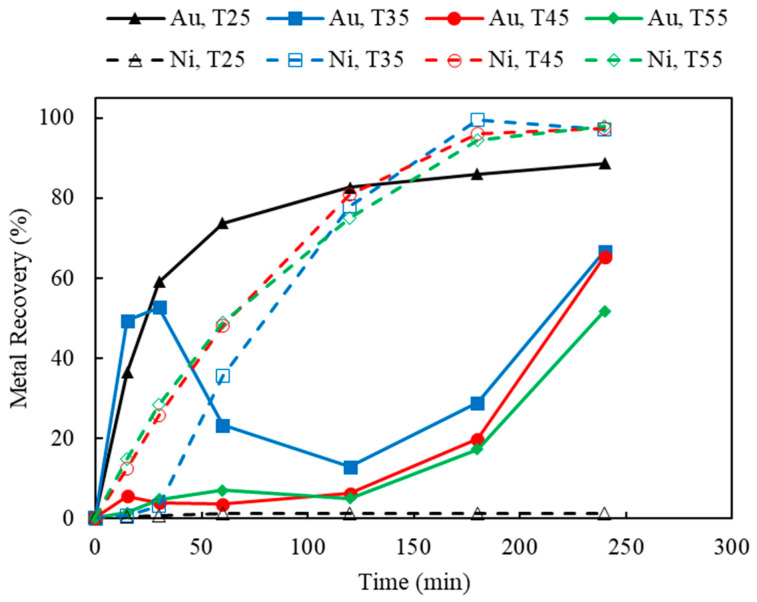
Au and Ni extraction (%) using AT/AH ratios of 0.25/0.5, 0.5/0.5, 0.75/0.5 (with fixed ammonia concentration of 0.5 M) (Experiment 9 and 17–19; S/L ratio: 10 g/L; particle size: −2 mm; Cu(II) 50 mM; aeration: 120 mL/min; Temperature: 25 °C).

**Figure 20 materials-16-04940-f020:**
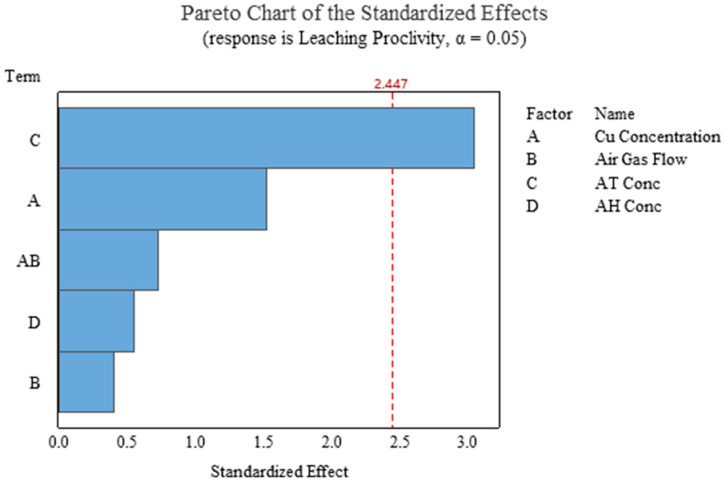
Pareto chart distinguishing significant factors contributing to leaching proclivity from insignificant factors (Experiment 2–4, 8–14).

**Table 1 materials-16-04940-t001:** Recovery % of Au from WPCBs using oxidative thiosulfate leaching.

Source of WPCBs	Lixiviant and Oxidizer	Pretreatment	Leaching Conditions	Au Recovery %	References
Various Waste Computers	(NH_4_)_2_S_2_O_3_ + CuSO_4_ + NH_4_OH	Physical separation, H_2_SO_4_ + H_2_O_2_	0.2 M (NH_4_)_2_S_2_O_3_, 0.4 M NH_4_OH, 0.02 M CuSO_4._	95.0	[36]
Waste Mobile Phones	Cu(II)–NH_3_–S_2_O_3_ solution	N/A	0.1–0.14 M S_2_O_3_, 0.2–0.3 M NH_3_, 15–20 mM Cu^2+^, pH from 10–10.5.	98.0	[39]
General Type	(NH_4_)_2_S_2_O_3_ + CuSO_4_·5H_2_O + NH_3_	H_2_SO_4_ + H_2_O_2_	0.5 M (NH_4_)_2_S_2_O_3_, 1 M NH_3_, 0.2 M CuSO_4_·5H_2_O, pH at 9.0.	98.0	[37]
Waste Mobile Phones	(NH_4_)_2_S_2_O_3_ + CuSO_4_	N/A	0.1 M (NH_4_)_2_S_2_O_3_, 40 mM CuSO_4_, pH from 10–10.5.	56.7	[40]
Waste Cell Phones	Na_2_S_2_O_3_ + NH_4_OH, with CuSO_4_ or H_2_O_2_	N/A	0.1 M Na_2_S_2_O_3_, 0.2 M NH_4_OH, 15–30 mM Cu(II), addition of H_2_O_2_, pH at 10–11.	15.0	[41]
	(NH_4_)_2_S_2_O_3_ + NH_4_OH, with CuSO_4_ or H_2_O_2_	N/A	0.1 M (NH_4_)_2_S_2_O_3_, 0.2 M NH_4_OH, 15–50 mM Cu(II), addition of H_2_O_2_, pH at 9–10.	2.0	
Discarded Mobile Phone	Cu(II)-NH_3_-Na_2_S_2_O_3_ solution, under N_2_ bubbling	N/A	0.1–0.14 M S_2_O_3_, 0.2–0.3 M NH_3_, 15–20 mM Cu(II).	91.0	[42]
Waste Cell Phones	(NH_4_)_2_S_2_O_3_ + NH_4_OH +Cu^2+^	H_2_SO_4_ + H_2_O_2_	0.08–0.12 M S_2_O_3_, 0.1–0.2 M NH_4_OH, 15 mM of Cu^2+^, pH at 10.5.	70.0	[38]
Waste Mobile Phones	Na_2_S_2_O_3_ + NH_4_OH + Cu^2+^	N/A	0.12 M Na_2_S_2_O_3_ + 0.2 M NH_4_OH + 20 mM Cu^2+^, pH at 10.	70.0	[43]
	(NH_4_)_2_S_2_O_3_ + NH_4_OH + Cu^2+^	N/A	0.12 M (NH_4_)_2_S_2_O_3_ + 0.2 M NH_4_OH + 20 mM Cu^2+^, pH at 10.	75.0	
Grinded E-waste Scrap	Na_2_S_2_O_3_/(NH_4_)_2_S_2_O_3_ + NH_3_ + Cu^2+^	Bioleaching	0.111 M S_2_O_3_^2−^ + 0.32 M NH_3_ + 30 mM Cu^2+^	87.0	[44]
Waste RAM Chips	(NH_4_)_2_S_2_O_3_+ NH_4_OH +Cu^2+^ with aeration	(NH_4_)_2_SO_4_ + NH_4_OH + Cu^2+^	0.25–0.75M (NH_4_)_2_S_2_O_3_, 0.5–1 M NH_4_OH, 0–75 mM Cu^2+^, 0–120 mL/min aeration, pH from 10–10.5.	>98.0	This study

**Table 2 materials-16-04940-t002:** Leaching parameters in Au-Cu(II)-NH_3_ thiosulfate system.

Pulp Density	Particle Top Size	c[Cu^2+^]	c[S_2_O_3_^2−^]	c[NH_3_]	Temperature	Aeration (21%O_2_)
g/L	mm	mM	M	M	°C	mL/min
10	3.4, 2.0, 1.2	25, 50, 75	0.25, 0.50, 0.75	0.50, 0.75, 1.00	25, 35, 45, 55	0 (Ar), 60, 120

**Table 3 materials-16-04940-t003:** Experimental ID and settings.

Experiment ID	Pretreatment	Particle Top Size	Temperature	c[Cu^2+^]	Aeration (21% O_2_)	c[S_2_O_3_^2−^]	c[NH_3_]	c[S_2_O_3_^2−^]/c[NH_3_]
		mm	°C	mM	mL/min	M	M	
1	No	2	25	75	120	0.5	0.5	1
2	Yes	2	25	25	0 (120 Ar)	0.5	0.5	1
3	Yes	2	25	50	0 (120 Ar)	0.5	0.5	1
4	Yes	2	25	75	0 (120 Ar)	0.5	0.5	1
5	Yes	2	25	25	60	0.5	0.5	1
6	Yes	2	25	50	60	0.5	0.5	1
7	Yes	2	25	75	60	0.5	0.5	1
8	Yes	2	25	25	120	0.5	0.5	1
9	Yes	2	25	50	120	0.5	0.5	1
10	Yes	2	25	75	120	0.5	0.5	1
11	Yes	2	25	50	120	0.25	0.5	0.5
12	Yes	2	25	50	120	0.75	0.5	1.5
13	Yes	2	25	50	120	0.5	0.75	0.67
14	Yes	2	25	50	120	0.5	1	0.5
15	Yes	3.4	25	50	120	0.5	0.5	1
16	Yes	1.2	25	50	120	0.5	0.5	1
17	Yes	2	35	50	120	0.5	0.5	1
18	Yes	2	45	50	120	0.5	0.5	1
19	Yes	2	55	50	120	0.5	0.5	1

**Table 4 materials-16-04940-t004:** Elemental assay of original Au finger (AF) as stamped and decopperized Au finger (AF) after Cu pre-leach.

Feed Materials	Ag	Al	Au	Co	Cu	Fe	Mg	Ni	Zn
	ppm	ppm	ppm	ppm	ppm	ppm	ppm	ppm	ppm
Original AF	93	26,331	6434	37	290,447	366	930	18,092	66
Decopperized AF	91	30,197	7713	41	28,984	697	926	19,163	66

**Table 5 materials-16-04940-t005:** Summary of experimental leaching results as presented in this study.

Experiment ID	Time To Au Drop (min)	Geometric Mean (min)	Time Weight Factor	Max Au % Recovery at Drop	Leaching Proclivity
1	0–15	7.5	32.0	0.0	0.0
2	60–120	90	2.7	102.6	273.5
3	60–120	90	2.7	71.3	190.1
4	180–240	210	1.1	60.4	69.0
5	120–180	150	1.6	100.0	160.0
6	>240	240	1.0	84.5	84.5
7	>240	240	1.0	80.3	80.3
8	120–180	150	1.6	94.7	151.5
9	>240	240	1.0	88.7	88.7
10	>240	240	1.0	79.8	79.8
11	>240	240	1.0	88.9	88.9
12	15–30	22.5	10.7	44.9	478.8
13	>240	240	1.0	97.9	97.9
14	180–240	210	1.1	101.7	116.3
15	180–240	210	1.1	86.9	99.3
16	180–240	210	1.1	81.0	92.5
17	15–30	22.5	10.7	49.4	526.8
18	0–15	7.5	32.0	0.0	0.0
19	0–15	7.5	32.0	0.0	0.0

**Table 6 materials-16-04940-t006:** Experiment 5–7 to Experiment 8–10 comparisons corresponding to 60 to 120 mL/min aeration rates (detailed coded coefficient table and Pareto chart obtained from Minitab are shown in the Appendix A).

Term	*p*-Value
Constant	0.009
c[Cu^2+^]	0.099
Aeration	0.946
c[Cu^2+^] × Aeration	0.891

**Table 7 materials-16-04940-t007:** Experiment 2–4 to Experiment 8–10 comparisons corresponding to 0 (Ar) to 120 mL/min aeration rates (detailed coded coefficient table and Pareto chart obtained from Minitab are shown in Appendix A and Appendix A).

Term	*p*-Value
Constant	0.003
c[Cu^2+^]	0.018
Aeration	0.045
c[Cu^2+^] × Aeration	0.073

**Table 8 materials-16-04940-t008:** *p*-Values obtained from Minitab as a result of factorial regression for all the impacting factors, excluding particle size, temperature, and 60 mL/min aeration.

Term	*p*-Value
Constant	0.027
Cu Concentration	0.177
Air Gas Flow	0.697
AT Conc	0.022
AH Conc	0.597
Cu Concentration × Air Gas Flow	0.490

## Data Availability

Available upon request.

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
