# Peer review of "Investigation of the Bimodal Leaching Response of RAM Chip Gold Fingers in Ammonia Thiosulfate Solution"

_materials, 2023, doi:10.3390/ma16144940_

Round 1
Reviewer 1 Report
1. The introduction is quite lengthy. The authors may revise again to make it more condense.
The explain of Au leaching mechanism can be move to section 2 (methods) or to section 3.
2. The results of metal composition after decopperization can be presented as the content of metals in the leach liquor (Cu, Ni…) and in the residue (Au…) for more comprehensive.
3. The SEM analysis of untreated AF may move to the section 2 (meterials and methods) as it is the characterization of the samples.
4. Section 3.3 is the effect leaching time; however, it included the results of SEM characterization of AF with and without aeration (line 350 to 390)?
5. The authors have presented the experiment design with 3 level, 2 factors (Ex 2-10), why didn’t estimate the regression based on this 32 design but divide to 2 times of 22? And the 22 design needs to repeat 3 times to have more reliable data of statistic from Minitab.
6. The results for effect of AT and AH shown table S7 and Figure S8 presented the regression of leaching depending on two factors? But the Figure 16 show the linear regression of only 1 factor. Can the authors explain for more details why those designs were selected to investigated?
7. Section 4, there were so many variables (4 factors), but the model shows only 1 factor has significant effect. It can be attribute to not enough number of experiments or the authors can check again the value of R-square. If the R-square value is too small, it means the regression model is not adequate to explain the influence of those factors on leaching efficiency.
Author Response
The authors express their gratitude for the reviewer's valuable feedback to enhance the manuscript's quality. Kindly refer to the attached file for a comprehensive response to the reviewer's comments.

Reviewer 2 Report
This article bring useful new insights to the community dealing with hydrometallurgy recovery of gold and its separation from Cu and Ni. And as such should be published once the following comment has been adressed:
there are so many RAM packaging possibility that the authours should specify in their materials and method, what kind of RAM packagesthey dealt with. Ideally pictures of the RAM should be given, at least in supplementary materials. Else, little reproducibility os to be expected.
Author Response

(The authors gave the same response as above.)

Reviewer 3 Report
The title of the manuscript, Investigation of the Bimodal Leaching Response of RAM Chip 2 Gold Fingers in Ammonia Thiosulfate Solution," is interesting. I have a few minor comments, after which it can be accepted for publication.
1. Authors clearly explain the abstract so that any reader can understand it.
2. Authors should improve the introduction and add recent references.
DOI: 10.1039/D1RA00338K https://doi.org/10.1016/j.esr.2019.100431
3. In the introduction, authors should show a strong interest in why recycling is important, with a focus on hazardous materials.
4 How do you manage chemical waste after recycling? explain
5. Methodology and results are well described.
6. The conclusion is also understandable.
Author Response

(The authors gave the same response as above.)

Round 2
Reviewer 1 Report
The authors has revised the section 2.2 by adding the leaching mechanism of Au, but it shows the same contents and equations in the Introduction.
Please check again and revise to avoid the repetition.
Author Response
The author sincerely appreciates the reviewer's carefulness in catching this mistake. The author meant to move Eqs.6-7 as a further discussion on the detailed Au leaching mechanism in session 2.2. But accidentally moved Eqs. 1-7 altogether. The repeated content has been deleted as shown in Lines 155-168 (revised version). The repetition in the text has also been checked elsewhere in the revised version.